# Autonomous Road Vehicles: Challenges for Urban Planning in European Cities

**Nikolaos Gavanas**

Directorate General for Research and Innovation (DG RTD), European Commission, 1049 Brussels, Belgium; nikolaos.gavanas@ec.europa.eu

**Abstract:** Autonomous vehicles will significantly affect mobility conditions in the future. The changes in mobility conditions are expected to have an impact on urban development and, more specifically, on location choices, land use organisation and infrastructure design. Nowadays, there is not enough data for a real-life assessment of this impact. Experts estimate that autonomous vehicles will be available for uptake in the next decade. Therefore, urban planners should consider the possible impacts from autonomous vehicles on cities and the future challenges for urban planning. In this context, the present paper focuses on the challenges from the implementation of autonomous road vehicles for passenger transport in European cities. The analysis is based on a systematic review of research and policy. The main outcome of the analysis is a set of challenges for urban planning regarding the features of urban development, the local and European policy priorities, the current lack of data for planning and the potential for autonomous vehicles to be used by planners as data sources. The paper concludes that tackling these challenges is essential for the full exploitation of the autonomous vehicles' potential to promote sustainable urban development.

**Keywords:** autonomous vehicle; connected and automated driving; sustainable urban development; urban planning; impact; challenge; Europe

---

## 1. Introduction

The World Economic Forum identifies three technological megatrends of the fourth industrial revolution (Industry 4.0), i.e., [1]: i. connectivity; ii. artificial intelligence; iii. flexible automation. The technology of the fully autonomous vehicle (AV) is aligned with these megatrends, especially in the case of connected and autonomous vehicles (CAV), which allow for the communication between vehicles, infrastructure and other road users (V2X connectivity) [2,3].

Scientists believe that the wide-scale implementation of AVs will bring transformative changes in mobility and accessibility, travel patterns, safety and security, energy efficiency, emissions, employment, data availability, governance and business models [4–8]. However, the scarcity of data for the real-life assessment of these changes often leads scientists to uncertain or even controversial conclusions. For example, the estimations regarding the possible impacts of AVs on green house gasses (GHGs) range from an 80% reduction to a threefold increase [9].

National and local governments will have to assess their transport strategy in view of the AV evolution [10]. Furthermore, planning will have to adapt to this uncertain future [11]. Taking into account that the purpose of the transport system is to provide access for people and goods to the locations where activity is conducted [12], it is evident that spatial planning and the organisation of land uses face their own challenges due to the implementation of AVs.

The objective of this paper is to present the possible impacts of AVs on cities in a comprehensive way and to outline the corresponding challenges for urban planning. In particular, the paper focuses on the challenges for urban planning from the wide-scale implementation of autonomous road vehicles

for passenger transport in European cities. The analysis refers to autonomous (or driverless) road vehicles of SAE's Level 5 of automation [13]. These vehicles may be privately owned passenger vehicles, shared-use passenger vehicles or public transport vehicles. In the case of shared-use or public transport CAVs, the vehicles can be enablers of mobility as a service (MaaS), i.e., door-to-door mobility based not on vehicle ownership, but on the integration of publically available transport services [14]. In the context of MaaS, door-to-door mobility is the seamless mobility of people and goods from the origin to the final destination of a journey using a single, integrated service [15].

The international literature does not yet sufficiently address the challenges for planning due to the implementation of AVs [16]. There is, however, a need to address these challenges in a timely manner, as experts estimate that fully autonomous road vehicles will be available for market uptake in the next decade and that they will comprise part of the vehicle fleet by 2050 [17–19]. Nowadays, urban planners argue about the ability of AVs to contribute to their planning objectives, while the cities of the future will probably have to find ways to adapt to the emergence of autonomous mobility, i.e., the mobility that is based on autonomous vehicles [20,21]. The thorough understanding and tackling of the challenges for urban planning is particularly relevant to Europe, where urban areas account for 70% of the population (with an estimation of an increase to 80% by 2050), 80% of the energy consumption and 85% of the gross domestic product (GDP) [22].

European cities present different features concerning the relation between urban development and transport in comparison to cities in other regions of the world. There are over 800 cities with more than 50,000 inhabitants in Europe. Approximately 85% of them have a population between 50,000 to 250,000 inhabitants. Only four cities in the European Union are included in the 80 most populated cities in the world. Asian and Latin American cities are in the top five of the specific list [23,24]. The public transport system plays a key role in the development of European cities since the 1800s. As a result, they are more compact and dense in comparison to the cities of North America and other regions of the world that relied on the private car to service their mobility needs [25]. The average population density of cities in Europe is 3000 inhabitants per km$^2$ and equal to the minimum density to sustain efficient public transport. The average density of cities in North America is almost half this density [26]. The relatively high density of European cities is often combined with mixed land uses, which relate to shorter daily trips and higher shares of public and active transport. Furthermore, many European cities date back to classical antiquity, the Roman Empire or the Middle Ages and thus, accommodate historical centres with limited roadway capacity and areas with restricted access to private cars. The present paper outlines the challenges for urban planning in the context of the above specific features in European cities. However, these challenges can be adjusted to the characteristics of urban areas in other regions of the world.

In order to outline the challenges for urban planning due to the implementation of AVs in European cities, the present paper addresses the following questions:

- What are the possible impacts from autonomous road vehicles on urban development?
- How is the concept of the autonomous road vehicle integrated into the policy priorities for sustainable urban development?
- What are the challenges linked to the lack of data about the impacts from autonomous road vehicles on urban development; what are the opportunities from the use of the autonomous road vehicles as data sources for urban planning?

The impacts from the wide-scale implementation of AVs on European cities depend on the features of each city and its transport system. They also depend on the urban development policy, either at the level of local priorities or at the level of common goals for the European Union and its member states. Furthermore, they depend on the specific type of AV to be implemented (AVs for private, shared and/or public transport, with or without V2X connectivity). Under these complex conditions, urban planners should combine different sources of data, exchange information and create strategic synergies in order to:

- Assess the possible impacts of autonomous vehicles on urban areas.
- Integrate autonomous mobility solutions to urban planning in order to support the specific needs of the examined city and to achieve the common goals for socio-economic and environmental sustainability.

## 2. Methodological Approach

The analysis of the present paper is based on the systematic review of research literature and policy documentation. The review was conducted in three parts:

- First, the review of research literature was directed towards the potential impacts of the autonomous road vehicles on urban mobility and accessibility conditions. Then, the possible effects of these impacts on the features of urban development were explored.
- Taking into account that planning is in line with specific policy goals, a review of policy documentation and relevant literature was conducted to investigate the relation between AVs and the policy priorities for urban development, with focus on sustainable development.
- A separate part of the literature review addressed the issue of data, both in terms of the absence of sufficient data for evidence-based planning today and in terms of the future contribution of AVs to big data, i.e., large data sets that are characterised by high volume, variety, velocity and veracity [27].

The findings from each part of the literature review led to the outline of specific challenges for urban planning. These challenges are summarised in tables at the end of each subsection and synthesised in the form of conclusive remarks.

## 3. Main Parameters and Related Challenges for Urban Planning

The main parameters that describe the possible effects on urban development from the implementation of AVs for urban passenger transport are presented in the following subsections. These parameters are related to specific challenges for urban planning in Europe.

### 3.1. Possible Impacts of Autonomous Road Vehicles on Urban Development

3.1.1. Value of Time, Accessibility and Location Choice

Autonomous road vehicles are expected to change the availability and quality of transport services. Driverless cars can increase the availability of road transport services by providing mobility for groups of the population that cannot drive conventional cars, as discussed below, but also by enhancing shared mobility [28]. Moreover, the AVs have the ability to affect the quality of transport services by decreasing accidents caused by human error and by making travelling more comfortable and efficient through smoother braking and finer speed adjustment [29]. These changes will not exclusively affect road transport but all modes of urban and interurban transport, due to their complementarity and competition with road transport. The impact from these changes depends on the use of AVs as privately owned vehicles, shared-use vehicles, public transport vehicles or a combination of the above categories [30].

The new and enhanced transport services are expected to affect the value of time and the perceived cost for the traveller. In the case of everyday passenger mobility, the value of time for commuting drivers and professionals who have to drive as part of their job (such as couriers and delivery services) will be highly affected, since the drivers will be able to replace the task of driving with other activities [31].

Moreover, driverless cars may become accessible to people that cannot drive conventional vehicles, such as disabled people, the elderly and underage travellers. In the case of shared use and public transport, AVs can also enhance the mobility choices of people who cannot afford to own a car [32].

The above-mentioned changes in the value of time, cost and access will affect the location choices of households and firms. Previous experience, such as the rise of private car use in the United States in

the first half of the 20[th] century, shows that the drop in the value of time and the increase in accessibility often lead to urban sprawl. Despite the fact that most European cities are more compact compared to cities of the United States, trends of significant urban sprawl have become apparent in Europe since the 1950s [33].

On the other hand, there are scientists who believe that the use of AVs in the context of a demand-responsive and flexible public transport service can enhance urbanisation [34,35]. This is particularly relevant to European cities, where urban development is closely linked to public transport. Therefore, urban planners should investigate the different ways that AVs can be implemented and the possible impacts on the location choices of households and firms, in order to choose the most suitable way to support urban development priorities.

### 3.1.2. Traffic, Parking Conditions and Land Use

Autonomous mobility will probably lead to more normalised flows, more stable speed profiles, less accidents and more uniform travel behaviour [36]. At the same time, it will increase accessibility, which is expected to eventually increase travel demand with possible impacts on congestion [37]. The impact on congestion is expected to be higher during the period of coexistence of AVs with conventional and partially autonomous vehicles, pedestrians, bicycles, e-scooters and other "light mobility" modes in dense urban environments [38]. This is due to the different features of the modes and the different individual travel choices of their users. It should be highlighted that the coexistence of different transport modes in dense urban environments is often observed in the historical centres of European cities.

From an engineering perspective, the autonomous vehicles require less headway distance and less lane width than conventional vehicles because of their high precision in driving. In this way, AVs can enhance the capacity of road infrastructure [39]. In the case of shared-use vehicles, which are expected to operate with relatively higher occupancy rates than private cars, the effect on capacity could be higher. The spare capacity will lead to the freeing of public space, which can be used to develop infrastructure for other modes, such as active transport modes, or for other land uses. The freeing of public space due to less requirements for roadway capacity comprises an opportunity for land use planning, especially in the dense central areas of European cities. Moreover, the ability of autonomous vehicles to move along segments with poor geometrical features may create new opportunities to better service the less accessible urban areas.

On the other hand, the high traffic density due to less headway distance and lane width may create problems for the safety and comfort of pedestrians and cyclists, especially in mixed traffic conditions [40]. The development of mixed traffic areas is currently a common practice for traffic calming in many neighborhoods of European cities [41,42].

In Europe, private cars are parked for about 95% of the time [43]. A share of approximately 30% of the traffic in city centres is due to drivers searching for free parking spaces [44]. Planners and policy makers in Europe aim for sustainable urban development schemes that consider the impacts of parking on the degradation of the urban environment [45]. In this context, the ability of AVs to park themselves at a distance from the destination of a trip, after dropping off their passengers, can lead to innovative approaches for the organisation and management of parking spaces. In addition, AVs will need less space within a car park, as both access and parking manoeuvres will be automated and highly precise. This will allow car parks to service the same number of AVs on less surface than the car parks for conventional vehicles [46,47]. Consequently, on-street parking demand at a short distance from the final trip destination is expected to decrease, freeing space for other land uses, while it is possible for on-street parking to be replaced by off-street car parks. This is particularly important for congested areas with low on-street parking supply and high demand during peak periods, such as commercial business districts (CBDs).

The proximity of car parks for AVs to the final destination of trips depends on the use of the AVs as private or shared-use vehicles. In the case of shared-use AVs, the passenger may use the closest

available vehicle, allowing for more flexibility in the allocation of parking spaces. On the other hand, the "empty" kilometres travelled by the AVs, i.e., the distance travelled by AVs without passengers, should also be considered in the planning for the allocation of parking spaces [48]. Thus, planners should evaluate the benefits from the reorganisation of the city's parking system and the opportunities for freeing space around highly attractive destinations against the externalities from the "empty" kilometres travelled by AVs.

### 3.1.3. Infrastructure, Networks and Design

The wide-scale implementation of AVs is expected to create new requirements and standards for the design of infrastructure in order to facilitate their navigation and to ensure the safety of other road users [49,50]. There is a potential to link these new design concepts with the overall transition towards an increasingly integrated, connected and "smart city". There are many definitions of "smart cities" in the literature. For the purpose of this paper, a "smart city" is defined as a city where "investments in human and social capital and traditional (transport) and modern (ICT) communication infrastructure fuel sustainable economic growth and a high quality of life, with a wise management of natural resources, through participatory governance" [51]. The potential of integrating the new design concepts related to AVs to the "smart" city infrastructure is of high significance for Europe, as the European Union actively supports the development of "smart" city infrastructure through policy measures in the framework of the Innovation Union flagship and other initiatives [52].

In the context of systems integration, European researchers have been exploring the potential of driverless electric vehicles for urban transport since 2010. Vehicle-to-grid (V2G) connectivity of electric vehicles (EVs) is tested to assess the potential for the fleet operators to provide ancillary services to the city's electrical system [53,54]. The integration of the infrastructure for AVs to the overall "smart" city infrastructure comprises a main challenge for urban planning.

Another requirement for the design of urban infrastructure relates to the pick-up and drop-off areas of AVs. These areas should be designed in a way that ensures the safety and comfort of the passengers and access to the adjacent land uses. The efficient design of these areas should take into account the prospect of making the AVs accessible to disabled people and the vulnerable users of the transport system. In this framework, some European cities will have to intensify their efforts to make their overall urban infrastructure accessible to all.

Table 1 presents the challenges for urban planning related to the possible impacts of autonomous road vehicles on urban development in Europe.

**Table 1.** Parameters of impacts of autonomous vehicles (AVs) for passenger road transport on urban development and relevant challenges for urban planning.

| Parameters | Challenges |
|---|---|
| Value of time, accessibility and location choice | • Appropriate combination of AV solutions as private, shared and/or public transport to contribute to the urban development goals. |
| Traffic and parking conditions and land use | • Opportunities for freeing public space and land use management in relation to the potential externalities of AVs. <br> • Potential contribution of AVs to service areas of limited roadway capacity and poor roadway features (e.g., historical centres). |
| Infrastructure, networks and design | • Opportunities for innovative design of urban infrastructure to facilitate AVs and other road users and to integrate the AV network into the energy and telecommunication networks. |

*3.2. Integration of the Autonomous Road Vehicle Concept with the Priorities for Sustainable Urban Development*

### 3.2.1. Policy Priorities at European Level

Since 2001, the European Union promotes sustainable development as the balanced development that meets the needs of the present without compromising the ability of future generations to meet

their own needs. Sustainable development is based on three pillars, i.e., the social, the economic and the environmental [55,56]. Recently, the European Union engaged to strengthen its efforts to achieve specific targets for sustainable development both internally, with the Energy Union [57], and globally, with its continuous contribution to the targets of the Paris Agreement and the Sustainable Development Goals (SDGs) [58,59].

The principles of sustainable development are evident in the priorities of the sectoral policies of the European Union, including transport policy. One of these principles refers to the contribution of technological innovation to the promotion of sustainability. A part of the Europe 2020 strategy, developed in 2010 in order to promote smart, sustainable and inclusive growth, acknowledges the significant impact of urban transport on congestion and emissions [60]. The Transport White Papers of 2001 and 2011 and the Green Paper on Urban Mobility highlight the importance of intelligent transport systems (ITS) and new technologies for the achievement of sustainable mobility in European cities [61–63]. In 2013, the European Union set the Europe-wide urban mobility planning guidelines, i.e., the Sustainable Urban Mobility Plans, and emphasised the need for innovative urban mobility solutions to reach the Europe 2020 targets [64].

Moreover, technological innovation in the context of "smart" cities is a priority of the agreement signed by the EU ministers responsible for urban matters in 2016 [65]. Regarding urban transport, the agreement focuses on the enhancement of connectivity and access for all. These priorities relate to the potential advantages of AVs regarding inclusiveness and the ability to reach less accessible areas, as discussed in the previous subsection.

In 2018, the European Commission published a communication on the strategy for the implementation of autonomous mobility. The communication mentions the potential contribution of AVs to urban planning due to their ability to foster shared mobility and to free public space [66].

Based on the above policy framework, the European Union promotes autonomous vehicles to strengthen sustainable urban mobility by implementing specific instruments and interventions. Representative examples comprise [67–70]:

- Projects funded by the European Union's Research and Innovation (R&I) Framework Programme, which aims at testing and assessing the impacts from Connected and Automated Driving (CAD).
- The WISE-ACT (Wider Impacts and Scenario evaluation of Autonomous and Connected Transport) action of the European Cooperation in Science and Technology (COST) organisation.
- The New Mobility Services (NMS) initiative of the European Innovation Partnership on Smart Cities and Communities (EIP-SCC).
- The Cooperative ITS (C-ITS) deployment and the C-Roads Platform under the Connecting Europe Facility (CEF).

At the national level, several European countries fund research projects, tests and demonstrations, explore regulatory aspects and set strategies and action plans for autonomous mobility, including Austria, Belgium, Czechia, Finland, France, Germany, Hungary, Latvia, the Netherlands, Poland, Portugal, Spain, Sweden and the United Kingdom [71].

From the above, it can be concluded that sustainable development is a common policy goal for European cities and that technological innovations, such as the AVs, can potentially contribute to this goal. In this context, urban planners should assess the possible contribution of AVs to the social, economic and environmental pillars of urban sustainability.

3.2.2. Differentiation of Policy Priorities at the Local Level

A main challenge in the setting of a common policy framework for sustainable development in Europe stems from the disparities between the different regions [72]. High heterogeneity is also observed between European cities, regarding their geographical characteristics, socio-economic features, development trends and governance structures [73]. This heterogeneity affects the characteristics and the speed in the adoption of innovation. For example, the differences between the peripheral and

central regions of Europe in the density of urban land use and in the availability and coverage of public transport are expected to play a key role in the adoption of AVs as public transport vehicles [74]. It can be concluded from the above that there cannot be a "one-size-fits-all" solution for European cities. The local characteristics and the strategic priorities of each city should be decisive factors in the planning process.

Table 2 presents the challenges for urban planning related to the potential integration of the autonomous road vehicle concept with the priorities for sustainable urban development.

**Table 2.** Parameters of potential integration of the AV concept with the sustainable urban development policy and relevant challenges for urban planning in Europe.

| Parameters | Challenges |
|---|---|
| Policy priorities at European level | • Consideration of the full extent of social, economic and environmental impacts of AVs on urban development. |
| Differentiation of priorities at local level | • Combination of local features, needs, trends and policy objectives for urban development with the overlying sustainable development policy framework, in order to select the appropriate solutions for the implementation of AVs. |

*3.3. Challenges Related to Data on the Impact of Autonomous Road Vehilces*

3.3.1. Lack of Data for Current Planning Purposes

As discussed in the previous subsections, the expected impacts from AVs on urban development expand beyond the effects on the urban form and the design of urban infrastructure and touch upon the social, economic and environmental pillars of sustainable urban development. However, the lack of data about the real-life impacts from the wide-scale implementation of AVs hinders the assessment of the expected changes.

In order to cope with this uncertainty, scientists often have to make simplifying assumptions in their assessments of the possible impacts of AVs. For example, the ability of conventional macroscopic models to assess the socio-economic and environmental impacts of AVs depends on assumptions about possible changes in travel choices and behaviours, due to unavailability of sufficient real-life data [75]. Taking into account the above-mentioned high heterogeneity between European cities, some of these assumptions may not be suitable for the assessment of impacts at the urban level [76]. Another approach is to synthesise conclusions based on:

- The implementation of full or partial automation in other transport modes, such as urban rail, and urban logistics. Rail automation has a long history in European cities with the first automatic train put in operation in 1967 in London. The evolution of rail automation and its impact on urban development can provide some indications about the possible changes due to the automation of the public bus system [77].

- The experience from past innovations in the road transport sector, such as the previously mentioned private automobile revolution. The differences in urban development during the second half of the 20th century between the automobile dependent cities of the United States and the public transport orientated cities of Western Europe [78] can help planners to assess the possible impacts of implementing AVs as private, shared or public transport modes.

- The lessons learnt from the automation in sectors other than the transport sector. An example in the field of socio-economic impacts refers to the introduction of automatic teller machines (ATMs) in the banking sector. According to a recent study, the implementation of ATMs led to the decrease of staff per bank branch but also to the opening of more branches due to less operating costs, affecting the relocation of activity but not the total number of employees [79]. Evidence and conclusions from such studies can help planners to anticipate changes due to the implementation

of AVs, provided that they take into account the differences and similarities between the examined sectors and technologies.

In addition, expert surveys and user acceptance surveys can provide insight into the expected changes [80]. Data can also derive from relevant research projects. In specific, the current European Union's R&I Framework Programme (H2020) is currently funding projects to collect data and information about the expected impacts from autonomous vehicles on urban mobility, which could be useful for future urban planning purposes [81]. As the technology of AVs is advancing and pilot testing is conducted in gradually more complex environments (including urban settings), larger amounts and wider coverage of data and information is collected [82]. Planners should stay up-to-date with the enhanced data and the new research findings.

Some cities are relatively advanced in "smart" mobility solutions, i.e., solutions based on new technologies for the optimisation of transport and the efficient use of all modes [83]. These cities are expected to adopt AVs at a faster rate than the rest [84]. They will also be the first to collect data about the implementation of AVs. Planners and planning authorities should promote the sharing of data and the exchange of information between the cities that are advanced and the ones that lag behind.

3.3.2. Enhanced Data for Future Planning Purposes

A fully operational AV on the urban road network will collect and process large amounts of data about the vehicle, the passenger, the traffic and the surroundings. If connected, an AV can continuously broadcast information about its position and speed, contributing to the assessment of traffic conditions, but also about travel patterns, road conditions, air quality, noise, etc. [85,86]. These data can be used for purposes beyond fleet and traffic management [87]. In the case of a connected and "smart" city, these data can be shared and combined with other big data deriving from the local government services, the operations of companies and the activities of citizens and visitors [88].

Urban planning authorities can take advantage of this potential, provided that they have access to the databases of autonomous vehicle fleet operators, traffic management authorities, vehicle manufacturers and ICT managers [89]. A relevant example is the public private partnership (PPP) between the US Department of Transportation and Sidewalk Labs (subsidiary of Google's parent company Alphabet) with the purpose of improving existing transport infrastructure in big cities through connected and autonomous mobility [90].

Data from AVs are linked with two issues for further consideration. The first issue concerns the potential threats to privacy, data protection and cyber security. The vulnerability of AVs against cyberattacks and data mismanagement increases with the level of connectivity and information exchange [91]. These threats will not be discussed in the present paper, as they are not exclusively related to planning. The second issue concerns the current trend of shifting the emphasis from long-term urban planning strategies to short-term planning adjustments. The availability of real-time or near real-time data enhances this trend [92]. However, strategic planning is still essential for European cities, due to their commitment to contribute towards the achievement of the long-term, common goals for sustainable development in the European Union, as discussed in the previous subsection.

Table 3 presents the challenges of the lack of data for current urban planning purposes and of the data availability for future planning purposes due to the implementation of autonomous road vehicles.

**Table 3.** Parameters regarding the current lack of data and the enhanced data availability in the future due to the implementation of AVs and relevant challenges for urban planning in Europe.

| Parameters | Challenges |
|---|---|
| Lack of data for current planning purposes | • Need to effectively combine existing data and information from various sources with ad hoc surveys.<br>• Data exchange between research and urban planning.<br>• Strategic synergies between European cities for the exchange of data and transfer of know-how. |
| Enhanced data for future planning purposes | • Use of AVs as data sources to monitor their impact on the city and support the overall urban planning process with big data. |

## 4. Conclusive Remarks

According to the literature, autonomous vehicles (AVs) are expected to be ready for market uptake in the next decade and to bring transformative changes in the mobility and accessibility conditions of urban areas. These changes will affect the features of urban development. More specifically, the wide-scale implementation of AVs will affect the location choices of households and firms, the availability of public space and the access to areas with poor roadway characteristics. This will create potential for the reorganisation of land uses. Furthermore, new opportunities are expected for the innovative design of urban infrastructure as well as for the integration of the AV network into the energy network, in the case of electric AVs, and to the telecommunication network, in the case of connected and automated driving (CAD). CAD is also related to the potential of using the AVs as sources of big data for urban planners.

The above impacts and opportunities will affect the social, economic and environmental pillars of sustainable urban development. In order for urban planners to better prepare for these changes, they should focus on the understanding of the relation between autonomous mobility and sustainable development and on the assessment of the specific impacts from AVs on their cities. In this way, they will be able to take full advantage of the potential benefits from autonomous mobility and improve the synergies between cities, AV developers and operators, data managers and the research community.

**Disclaimer:** The information and views set out in this article are those of the author and do not necessarily reflect the official opinion of the European Union. Neither the European Union institutions and bodies nor any person acting on their behalf may be held responsible for the use which may be made of the information contained therein.

**Funding:** This research received no external funding.

**Conflicts of Interest:** The authors declare no conflict of interest.

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
