# Peer review of "Autonomous Road Vehicles: Challenges for Urban Planning in European Cities"

_urbansci, doi:10.3390/urbansci3020061_

Reviewer 1 Report

The paper provides an overview of the research conducted mainly in the last five years about urban mobility in relation to the advent of autonomous road vehicles in European cities. The Author’s aim is to list the possible impacts of autonomous vehicles (AVs) on cities and to highlight the consequent challenges that will present themselves to urban planners.

The analysis is divided in three main sections, each of which is focused on a specific topic: impact of passenger road transport on urban development and planning; impact of integration of AVs on sustainable urban development; impact of the current lack of data and of the possible future availability of these on urban planning.

The work is potentially fascinating, however, despite the subject presented in the Introduction and the large number of references provided, the various topics are not discussed in depth, nor are there any particularly original reflections or comments. It seems that the Author has mostly summarized a series of concepts that are nowadays fairly known, without adding something more. Besides, keywords that are very popular these days (smart, big data, ...) are easily used, but the basic concepts are not provided. I would suggest avoiding this procedure and to better explain or justify the following statements:

·         Line 104: which will be the main changes and how will they affect the quality of transport services?

·        Line 113-114: based on what the Author claims that AVs will be accessible to people who do not have a driving license?

·         Line 129-131: it would be useful to cite some references;

·         Line 152: the first sentence would need a reference;

·        The term "smart city" is often used within the text, but it is never explained what is meant by "smart"; the same applies to the expression "smart mobility solution".

In addition, please consider the following points:

·         Line 48: could the Author clarify the expression “…door-to-door mobility…”?

·       Line 84-94: since it is said that the work is divided into three parts (Line 83), a list of only three points would be clearer for the reader to get an overview of the work;

·         Line 315: it is preferable to repeat the full name in the conclusions before using the acronym to facilitate reading and for sake of clarity;

·         Line 323: the acronym should be added after the full name before using it;

·         For many of the references available online, the provided link produces an error, or the page is not found (see Refs. 1, 20, 29, 45, 62, 65, 66).

Author Response

Dear Reviewer,

Thank you for your useful comments. They were valuable to improve the quality of the paper. Your general comments were addressed by enriching the text with definitions, avoiding oversimplified use of terms, adding more examples from worldwide research and further analysing some thoughts. These changes were integrated without diverging from the original scope of the paper.

Your specific comments were addressed, as you may see below:

Line 104. The main changes and effects were addressed.

Line 113-114. The phrase referred to the potential use of AVs by underage travellers, who do not have a driving licence. However, your comment is right, as this remark is neither supported by literature nor needed to support the argument. Thus, it has been erased.

Line 129-131. Relevant references cited.

Line 152. Relevant reference cited.

Definitions and references of smart city and smart mobility were added.

Line 48. The expression “door-to-door mobility” was clarified by the addition of an explanatory sentence with citation.

Line 84-94. The comment is valid. The second point has been integrated to the first, since they comprise 2 sections of the first part of the literature review aiming at investigating the impacts on urban development.

Line 315. Valid comment. The full name was added.

Line 323. Another valid comment. The acronym was added.

Regarding the issue with internet links in the reference list, the links were re-tested from a private device with no special access rights and they seem to work. The sources were checked and found trustworthy (international organisations, consultancies, official forums, project websites, scientifc journals etc.). No errors appear. However, a change in one of the sites (reference number 14 in the revised document) was observed and the reference was replaced with a valid one. 

According to the journal’s instructions, you may find the changes and additions in the revised version of the paper in track changes.

Thank you once more for the time and effort you dedicated to improve this work.  

Respectfully,

The Corresponding Author

Reviewer 2 Report

This paper presents a summary of potential challenges on urban planning from the implementation of autonomous road vehicles (ARVs) for passenger transport. The analysis is based on review of research and policy.

The analysis is composed of three parts: potential impacts of ARVs on urban planning, integration of ARV concept for sustainable urban development, relevant data on studying the impact of ARVs.

The reviewer has two suggestions:

1. What are the differences between European cities and US cities or China cities? It seems that all the potential challenges are universal.

2. The third part on data is a more interesting topic. However, this paper only provides a brief introduction. More elaborations on the statements are preferred.

Author Response

Dear reviewer,

Thank you for your useful comments. They helped to enhance the content and clarify the scope of the paper.

As you have accurately commented, the outlined challenges apply to many regions of the world that will implement AVs and prioritise sustainable development. The paper presents specific arguments to identify the challenges, examples for stressing the importance to tackle these challenges and a policy background which are all Europe-oriented. In this way, the discussion on these general challenges becomes more comprehensive and more relevant to the special features of this part of the world. However, the framework of challenges for a policy-oriented, evidence-based planning approach can be adjusted to other regions as well. In order to address your valid comment in the text, an effort was made to clarify the main features of the European cities in relation to the characteristics of urban mobility and urban development. The differentiations between these cities and other cities of other regions in the world are highlighted.This section was added in the corresponding part of the introduction.

Moreover, the aspect of data (current lack of data on impacts and potential for additional data to support planning) was elaborated more, as you suggested. In specific:

·       The threat of over simplifying assumptions was further explained with an example from literature.

·       Examples were added for the possible ways to combine data from automation in other modes and sectors and also for the experience from previous road transport “revolutions”.

·       The potential synergy between planning and research is discussed in view of gradually increasing data input from the latter about impacts of AVs in complex environments.

·       The possible data that can be collected and transmitted by an AV is presented to demonstrate that they can be useful for urban planning.

·       The subject of cyber-security and privacy is correlated with the technological advancement of connectivity and information exchange.

According to the journal’s instructions, you may find the changes and additions in the revised version of the paper in track changes.

Thank you once more for the time and effort you dedicated to improve this work. 

Respectfully,

The Corresponding Author

Reviewer 3 Report

Dear editor and dear author,

Thank you for giving me the exciting opportunity to review the present manuscript for a truly emerging journal like Urban Science and thank you for exposing me to an interesting research study respectively.

Autonomous vehicles (AVs) is arguably the next big thing in transport provision and their policy, planning and implementation dimensions and implications are a key issue that research has to address pro-actively. There is a plethora of research on this thematic agenda but there is still a lot of room for good contributions which adopt an applied social science approach and put forward conceptual frameworks useful for academics, practitioners and policy decision-makers. The challenges and opportunities affecting and reflecting the implementation of AVs and the way with which this new driverless reality can fit into current policy frameworks is a particular angle that make this paper a solid submission.

 As a whole, the paper is based on a fine idea, organised systematically and has the potential to eventually develop into a robust research contribution. The rhetoric used is very good and well-justified and there is consistent flow. It is a paper easily accessible to non-academic audiences too. I like it.

 Since the paper is a literature review-based one, the only minor amendment that I would like to suggest is the inclusion of a few more papers in the author’s bibliography. There is an abundance of papers lately on AVs so this recommendation will help significantly the author’s narrative to become stronger and clearer; the paper will be more representative of the emerging literature. 

I would advise the author therefore to read, evaluate and include in his revised manuscript the following papers. Every paper of this lot would be able to offer him at least an additional point each. You do not have to write much on them but there is a great amount of useful information very relevant to the existing paper:

 1.     Bansal, P. and Kockelman, K. M. (2017). Forecasting Americans’ long-term adoption of connected and autonomous vehicle technologies. Transportation Research Part A: Policy and Practice, 95, 49-63.

2.     Fagnant, D. J. and Kockelman, K. (2015). Preparing a nation for autonomous vehicles: opportunities, barriers and policy recommendations. Transportation Research Part A: Policy and Practice, 77, 167-181.

3.     Fraedrich, E., Heinrichs, D., Bahamonde-Birke, F. J., & Cyganski, R. (2019). Autonomous driving, the built environment and policy implications. Transportation research part A: policy and practice, 122, 162-172.

4.     Nikitas, A., Njoya, E. T., & Dani, S. (2019). Examining the myths of connected and autonomous vehicles: analysing the pathway to a driverless mobility paradigm. International Journal of Automotive Technology and Management, 19(1-2), 10-30.

5.     Parkin, J., Clark, B., Clayton, W., Ricci, M., and Parkhurst, G. (2018). Autonomous vehicle interactions in the urban street environment: A research agenda. Proceedings of the Institution of Civil Engineers: Municipal Engineer, 171(1), 15-25.

6.     Thomopoulos, N., & Givoni, M. (2015). The autonomous car—a blessing or a curse for the future of low carbon mobility? An exploration of likely vs. desirable outcomes. European Journal of Futures Research, 3(1), 14.

If this sole suggestion is dealt with appropriately I think that this would be a great research output.

Author Response

Dear reviewer,

Thank you for your useful comments, which helped to enhance the content of the paper. The proposed references contributed to the better understanding of many issues that the paper touches upon. Thus, all proposed references were integrated to the text with the appropriate additions and changes in the paper.

According to the journal’s instructions, you may find the changes and additions in the revised version of the paper in track changes.

Thank you once more for the time and effort you dedicated to improve this work. 

Respectfully,

The Corresponding Author

Round  2

Reviewer 1 Report

I thank the Author for having addressed and discussed the points raised during the first review. The work has been enriched and the overall quality of the paper has improved. Unfortunately, I still encountered some glitches with links to online references. However, when such errors occur, if the link is copied and pasted manually, then it works. Apart from this, I believe the paper provides an interesting collection of sources and information, and I think it can be accepted after minor text editing corrections.

Author Response

Dear reviewer,

I would like to sincerely thank you once again for your comments. Apologies for not being able to further address the issue of online references. According to your suggestion, the paper was revised and editing corrections were made.

Respectfully,

The corresponding author